# Evaluation Model Research of Coal Mine Intelligent Construction Based on FDEMATEL-ANP

Lin He [1], Dongliang Yuan [1,*], Lianwei Ren [1], Ming Huang [2], Wenyu Zhang [3] and Jie Tan [4]

1   School of Civil Engineering, Henan Polytechnic University, Jiaozuo 454003, China
2   China Construction Eighth Bureau Technology Construction Co., Ltd., Shanghai 200433, China
3   The Third Construction Engineering Co., Ltd. of China Construction Second Engineering Bureau, Beijing 100070, China
4   China Construction Fifth Bureau South China Construction Co., Ltd., Shenzhen 518000, China
*   Correspondence: ydlwyf@163.com

**Abstract:** To improve intelligent construction standard systems in coal mines, we must promote the high-quality development of the coal mining industry. The current intelligent construction of coal mines is inefficient. Considering the complexity and diversity of coal mine intelligent construction index factors, this paper proposes an intelligent coal mine construction evaluation model that integrates the fuzzy decision-making trial and evaluation laboratory (FDEMATEL) and the analytical network process (ANP). Firstly, the evaluation index system is established based on the intelligent construction of coal mines. Secondly, the FDEMATEL is applied to deal with the fuzziness in the evaluation process and determine the influence relationship between the evaluation indexes of coal mine intelligent construction to draw the ANP network structure diagram. Finally, super decision software is used to calculate the weight of coal mine intelligent construction evaluation indexes, and then obtain the combination weight and correlation degree of each evaluation index. By applying the evaluation model to conduct a comprehensive evaluation of coal mine intelligent construction, the results show that there is a significant correlation between the indexes affecting the intelligent construction of coal mines. Basic platform intelligence and safety monitoring intelligence are the two most important aspects of intelligent coal mine construction. Database construction, mobile internet construction, big data support, and model algorithm support are the key indexes affecting the intelligent construction of coal mines.

**Keywords:** intelligent coal mine; fuzzy mathematics; DEMATEL; ANP; index system; evaluation model

## 1. Introduction

Mineral resources are the important material basis for human survival, economic construction, and social development, among which coal has long been an important organizational part of China's energy structure [1,2]. According to the energy consumption records published by the National Bureau of Statistics, the annual consumption of coal in China has been stable above 3.5 billion tons in recent years, including approximately 3.99 billion tons in 2015, 3.88 billion tons in 2016, 3.91 billion tons in 2017, 3.97 billion tons in 2018, and 4.02 billion tons in 2019. During this five-year period from 2017 to 2021, the share of coal in total energy consumption has always remained above 56%, although it is gradually decreasing. The uncontrolled mining of coal resources has caused serious damage to the ecological environment and has affected the balance between resource exploitation and environmental protection. Therefore, the adoption of intelligent technology to drive the transformation and improvement of the traditional mining industry is necessary, promoting safe, efficient, economic, green, and sustainable development of traditional mining, which is new theme of future coal mine construction.

At present, with the deep integration of industrialization and information technology, emerging development models, such as intelligent cities, intelligent logistics, and intelligent

agriculture, have been derived [3]. Scholars at home and abroad have carried out in-depth exploration of these development models and achieved fruitful research results, but research regarding the intelligent construction of coal mines is still in its primary stages. Wei et al. [4] proposed an infrared thermal imaging personnel detection method to ensure the safety of coal miners, in view of the problem of weak light in intelligent coal mines on the general mining face. Zhang et al. [5], Fu et al. [3], Luo et al. [6], and He et al. [7] elaborated the basic theory, general framework, core issues, and control concept of intelligent mine construction from the perspective of key technologies in intelligent mines in China. Li and Zhan [8] used an underground metal mine in China as an example and tested the equipment, technology, and infrastructure platforms of this intelligent mine, and the tests showed the important role of intelligent technology to enhance mining. Wang and Huang [9] introduced the intelligent mining equipment developed by China during the 12th Five-Year Plan, which can meet the needs of China's mining industry. By comparing this with the relevant foreign equipment, it is believed that the domestic technology has reached the international level. Bing et al. [10] proposed an Intelligent architecture based on edge computing and used an example to verify that the architecture enables intelligent prediction of mine safety. Wang et al. [11] and Bai et al. [12] elaborated on the bottlenecks and challenges facing the construction of intelligent mines in China, including construction specifications, technical standards, data acquisition, terminal sensors, key technologies, risk management, and control capabilities. Chen and Wang [13] verified the good reliability and stability of wireless sensors applied in coal mine safety intelligent monitoring systems through simulation tests. Wo et al. [14] collected the policy documents released by the central and local governments on intelligent coal mine construction and explored the development trend of intelligent mine policies using text analysis. Lu et al. [15] used numerical analysis and MATLAB software to establish the mathematical model of coal mine gas accidents and realized the intelligent prediction of gas accidents through testing and analysis.

In terms of content, the above studies mainly focus on the overview of the development prospects, key technologies, and construction paths of intelligent mine construction. Only some scholars have provided in-depth analysis of intelligent coal mine safety monitoring and intelligent mining equipment. The above research results further enrich the basic theory of intelligent coal mine construction and have a certain guiding effect on the research of this paper. Therefore, this study takes a new coal mine in Shanxi Province as an example and uses the literature review, expert interviews, and questionnaires to extract the evaluation indexes of coal mine intelligent construction. Then, the evaluation index system of coal mine intelligent construction is constructed, and the evaluation model of FDEMATEL-ANP is used to elucidate the relationship between the indexes.

This paper is organized as follows. Section 2 presents the literature review related to the intelligent construction of coal mines. Section 3 introduces the research methods of this paper. Section 4 analyzes the data and obtains the calculation results. Section 5 discusses the results of the study. Section 6 summarizes the conclusions of this study.

## 2. Literature Review

Under the background of the rapid development of intelligent technology and equipment, intelligent construction of coal mines has become the only way for mining enterprises to realize the transformation, improvement, and high-quality development of the mining industry [16]. The intelligent construction of coal mines uses technologies such as big data, artificial intelligence, cloud computing, and sensor networks to improve production efficiency, ensure production safety, and rationalize resource deployment. Studies have shown that scientific evaluation index systems and standards can help coal mining enterprises clearly recognize the degree of their own coal mine construction intelligence and reduce the construction process to carry out projects of low practicality, thereby promoting the improvement of enterprise production efficiency [17]. Therefore, in order to accurately grasp the development direction of intelligent coal mine construction, improve

the standard system of intelligent coal mine construction, and promote the high-quality development of intelligent coal mine construction, it is important to carry out an evaluation of intelligent coal mine construction to determine the key factors affecting intelligent coal mine construction.

Li et al. [18] established an evaluation index system based on the "eight systems" of intelligent mine production, applied the AHP method to determine the weight of each index, and established an evaluation model based on the uncertainty measurement theory. Qiu and Tan [17] established an index system to measure the degree of intelligent mine construction by sorting out the logical system of intelligent mine construction. Guo [19] constructed an evaluation index system and grading criteria based on the stages of intelligent construction of open pit mines and evaluated the intelligence of open pit coal mines. According to the definition of intelligent mines, He [20] constructed the intelligent mine evaluation system and the grading evaluation method. The above literature has undertaken useful exploration on the evaluation of the intelligent construction of coal mines and formed a systematic theoretical basis. However, due to the lack of a unified standard systems for the intelligent construction of coal mines, the evaluation process is often influenced by the different knowledge levels, experience, and subjectivity of researchers, so the use of traditional research methods has been unable to achieve a scientific and accurate analysis. In view of this, research methods and tools in the field of complex science, such as analytic network processes (ANP) [21], structural equation modeling (SEM) [22], multi-agent model [23], data mining [24], interpretative structural modeling (ISM) [25], and the decision-making trial and evaluation laboratory (DEMATEL) [26], should be considered comprehensively.

At present, in the field of mining and energy research, some scholars have used these methods to complete the subject research. For example, Li et al. [27] applied a combined grey DEMATEL and ANP approach to evaluate the effective method of green mining performance in underground gold mines. Mardani et al. [28] provided a systematic review of articles on the application of MCDM methods in sustainable and renewable energy by reviewing the literature in several advanced journals. Rakhmangulov et al. [29] proposed a MCDM method using the combined fuzzy AHP- MARCOS to assess the sustainability of open pit mining systems. Lenarczyk et al. [30] applied the MCDM to assess which renewable energy sources should be used to generate electricity in Poland under current socio-economic conditions. Wang et al. [31] used fuzzy theory, DEMATEL, and MCDM to explore the key influencing factors of miners' unsafe behaviors in intelligent mines in order to improve the safe production environment of intelligent mines. Chen et al. [32] used the AHP-DEMATEL model to analyze the main factors for the occurrence of unsafe conditions of employees in the coal mine production process. Zhang et al. [33] combined ANP with the matter–element extension model to evaluate the factors influencing the adaptability of coal mine intelligent mining workings and to determine its adaptability level. Li et al. [34] used the entropy weight method and extension theory to construct an evaluation model for the safety of the intelligent mine working face.

By analyzing the existing research results, it is found that only a few scholars have been involved in studies on the intelligent construction of coal mines. The study of intelligent construction in coal mines still needs to be further developed. To supplement the theoretical research in this area, this paper establishes the intelligent construction evaluation index system of coal mines according to the content and characteristics of the intelligent construction of coal mines combined with relevant policies, through a literature review, expert interview methods, and the questionnaire survey method. Based on the complexity and diversity of evaluation indexes and the fuzzy randomness of the evaluation process, the advantages and disadvantages of each research method are fully considered. This paper proposes to use the FDEMATEL method to determine the influence relationship between each evaluation index, the ANP method to determine the index weight, and then build an evaluation model of coal mine intelligent construction based on FDEMATEL-ANP. Finally, the mixed weights and correlation ranking of indexes are derived to determine the key indexes of coal mine intelligent construction. On this basis,

suggestions are made to promote the intelligent construction of coal mines. It is expected that the research results will help the coal mining industry and enterprises to improve coal mine intelligence construction standards and accelerate the process of coal mine intelligent construction. At the same time, we describe the limitations of the current study and future research directions.

## 3. Materials and Methods

### 3.1. Intelligent Coal Mine Construction Connotations

Based on the deep integration of new generation information technology and traditional mining development technology, the intelligent construction of coal mines mainly uses advanced technology, equipment, and management methods to achieve active perception, automatic analysis, dynamic prediction, and collaborative control of mine production process, equipment working conditions, and environmental safety. According to the optimal decision model formed by intelligent analysis, the mine "human, machine, environment and management" operate in a highly coordinated unity, thus improving the equipment utilization rate and resource allocation rate, realizing the safety, efficiency, and sustainability of mine design, production, and operation management [1,7,35,36]. The intelligent construction of coal mines is a complex system of engineering; when the mine geological environment, production requirements, and mining status are different, the levels and standards to be followed are not the same, but mainly includes the intelligent construction of production, safety, management, and mining machinery [37].

### 3.2. Coal Mine Intelligent Construction Evaluation Index System

The evaluation index system of the intelligent construction of coal mines is an evaluation standard to measure the intelligent degree of mine construction and determine the goal of intelligent construction of coal mines. At present, there are differences in the foundation of intelligent construction of coal mines, and there is a lack of unified standards for guidance and constraints, resulting in a serious "information silos" phenomenon. According to the current situation of the development of intelligent coal mine construction, this paper starts from the content and characteristics of intelligent coal mine construction, based on the existing research results and the intelligent coal mine construction-related policy documents issued by national and local governments, such as "Guidelines on Accelerating the Development of Intelligent Coal Mines," "Guide to the Construction of Intelligent Factories (Mines) in the Non-ferrous Metals Industry (for Trial Implementation)," "Guide to the Construction of Intelligent Coal Mines (2021 Edition)," and so on. Adhering to the principles of scientific measurability, comprehensive and systematic index establishment, and being goal-oriented, the brainstorming method is used to organize an expert meeting to summarize the important indexes of intelligent coal mine construction evaluation. Finally, by integrating relevant policy specifications, expert opinions, and research results, the intelligent construction of coal mines is divided into basic platform intelligence, production process intelligence, safety monitoring intelligence, information management intelligence, and green development intelligence, a total of 5 first-grade indexes and 24 second-grade indexes (see Table 1). Among them, the basic platform intelligence and production process intelligence draw on literature [1,35]; safety monitoring intelligence and information management intelligence reference [38,39]; and green development intelligence draws on [20,36].

This paper takes a new coal mine in Shanxi Province as an example for the empirical study. This coal mine has a design production capacity of 15 million tons per year and is divided into the east mining area and west mining area. The east mining area is approximately 770 m long and 350~704 m wide, with an area of 381,600 square meters. The west mining area is 720 m long and 660 m wide, covering an area of 435,700 square meters. The recoverable reserves of the mine are 181.24 million tons as of now, with a service life of 13.9 years. The necessity of intelligent coal mine construction was determined by comparing the main technical and economic indexes of intelligent mining and traditional

mining in terms of mining process, main equipment of the mine, project organization structure, and technical personnel configuration.

**Table 1.** Evaluation index system for the intelligent construction of coal mines.

| Goal Layer | First-Grade Indexes | Second-Grade Indexes |
|---|---|---|
| Coal mine intelligent construction evaluation $A$ | Basic platform intelligence $I_1$ | Database construction $F_{11}$<br>Big data support $F_{12}$<br>Model algorithm support $F_{13}$<br>Mobile internet construction $F_{14}$<br>Information safety system construction $F_{15}$ |
| | Production process intelligence $I_2$ | Reliability of working surface equipment $F_{21}$<br>Intelligent diagnosis technology of equipment fault $F_{22}$<br>Mining collaborative design $F_{23}$<br>Production closed-loop control $F_{24}$<br>Safety prevention closed-loop decision $F_{25}$ |
| | Safety monitoring intelligence $I_3$ | Geological monitoring $F_{31}$<br>Ventilation and fire safety monitoring $F_{32}$<br>Electrical equipment safety monitoring $F_{33}$<br>Personnel safety monitoring $F_{34}$<br>Emergency rescue control $F_{35}$ |
| | Information management intelligence $I_4$ | Information collection coverage capability $F_{41}$<br>Data resource mining capability $F_{42}$<br>Data statistical analysis ability $F_{43}$<br>Information management system integration capability $F_{44}$ |
| | Green development intelligence $I_5$ | Intelligent dust control $F_{51}$<br>Intelligent control of toxic and hazardous substances $F_{52}$<br>Drained water reuse $F_{53}$<br>Clean energy utilization $F_{54}$<br>Ecological restoration management $F_{55}$ |

This paper calculates the influence degree of each evaluation index of coal mine intelligent construction this the FDEMATEL method and determines the network structure between the indexes. The ANP method is then used to determine the relative importance weights of each index. Finally, by combining the correlation degree, weight, and network structure diagram, the indexes are ranked and classified according to different attributes and importance weights to find the key indexes to promote the intelligent construction of coal mines. The combination of FDEMATEL and ANP not only solves the problem that the DEMATEL method treats all indexes as equal weights, but also considers the AHP method that ignores the mutual influence relationship between indexes, which can improve the objectivity of the calculation of index weight. Following comprehensive analysis of the above, this paper constructs a coal mine intelligent construction evaluation model based on FDEMATEL-ANP as shown in Figure 1, and the specific principles are as follows.

### 3.3. FDEMATEL Model

The DEMATEL method was first proposed by A. Gabus and E. Fontela, scholars at the Battelle Research Center in the United States, as a systematic methodology for studying and solving more complex and difficult problems using graph theory and matrix tools [40,41]. Firstly, the method uses the knowledge and experience of experts to analyze the logical relationships between indexes and transforms the complex relationships between indexes into easily understandable directed graphs and direct impact matrices. Secondly, based on the logical relationship of indexes, the method can calculate the influence degree, influenced

degree, centrality degree, and cause degree of each index, determine the causal relationship between indexes, and clarify the network connection of each index and its position in the whole system, so as to provide the basis for the construction of the network hierarchy model. However, considering that the DAMATEL method is based on experts' scores, the difference of experts' subjective judgments and the ambiguity of indexes influencing each other can have a large impact on the study results. Therefore, the FDEMATEL method, which combines fuzzy set theory with DEMATEL, is chosen [42]. The fuzzification of the direct influence matrix is achieved by converting the expert semantic evaluations into the corresponding triangular fuzzy numbers, and then the fuzzy numbers are converted into exact values by the CFCS (converting fuzzy numbers into crisp scores) defuzzification method proposed by Opricovic et al. [43]. The specific steps are as follows.

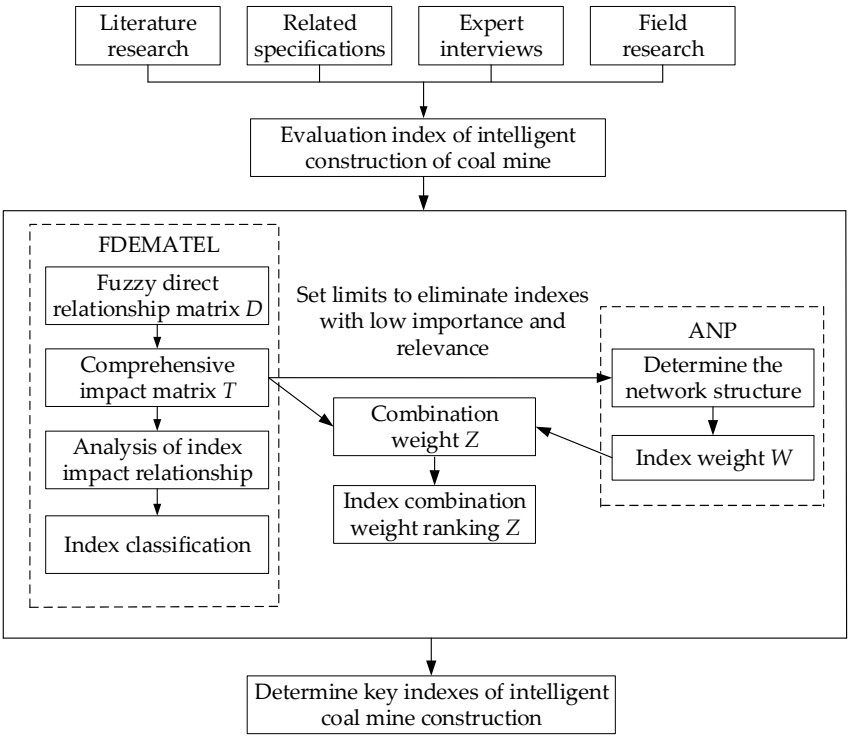

**Figure 1.** Evaluation model of coal mine intelligent construction based on FDEMATEL-ANP.

Step 1: determine the factor set and construct the language evaluation set. The degree of influence among the factors is classified as very low influence "1" low influence "2" medium influence "3" high influence "4" and very high influence "5" and then experts are invited to judge the relationship between the influence factors. The semantic conversion is shown in Table 2.

**Table 2.** Semantic conversion table [44].

| Linguistic Variable | Expert Ratings | Triangular Fuzzy Number |
|---|---|---|
| Very low influence (VL) | 1 | (0, 0, 0.25) |
| Low influence (L) | 2 | (0, 0.25, 0.5) |
| Medium influence (M) | 3 | (0.25, 0.5, 0.75) |
| High influence (H) | 4 | (0.5, 0.75, 1) |
| Very high influence (VH) | 5 | (0.75, 1, 1) |

Step 2: create the initial direct impact matrix $R$. The expert evaluation opinions are transformed into the corresponding triangular fuzzy numbers according to the expert linguistic evaluation set. The triangular fuzzy number $X$ can be represented by $(l, m, r)$. Assume that $X_{ij}^k = (l_{ij}^k, m_{ij}^k, r_{ij}^k)$ represents the triangular fuzzy evaluation of factor $i$ on

factor $j$ given by expert $k$, where $l$ is the conservative value (lower bound of the fuzzy number), $m$ is the estimated value, $r$ is the optimistic value (upper bound of the fuzzy number), and $l \leq m \leq r$.

Step 3: initial matrix clarification. Defuzzification using the CFCS method. The specific steps are as follows.

(1)　Standardized triangular fuzzy numbers.

$$
\begin{aligned}
a_{ij}^k &= (l_{ij}^k - \min l_{ij}^k) / \Delta_{\min}^{\max} \\
b_{ij}^k &= (m_{ij}^k - \min m_{ij}^k) / \Delta_{\min}^{\max} \\
c_{ij}^k &= (r_{ij}^k - \min r_{ij}^k) / \Delta_{\min}^{\max} \\
\Delta_{\min}^{\max} &= \max r_{ij}^k - \min l_{ij}^k
\end{aligned}
\tag{1}
$$

where $a_{ij}^k$, $b_{ij}^k$ and $c_{ij}^k$ represent the normalized left, middle, and right values, respectively. $\Delta_{\min}^{\max}$ represents the difference value between the right-side value and the left-side value.

(2)　Normalized left and right values.

$$
u_{ij}^k = b_{ij}^k / (1 + b_{ij}^k - a_{ij}^k) \tag{2}
$$

$$
v_{ij}^k = c_{ij}^k / (1 + c_{ij}^k - b_{ij}^k) \tag{3}
$$

where, $u_{ij}^k$ and $v_{ij}^k$ represent the normalized values of the left and right values, respectively.

(3)　Calculate the total standardized value $y_{ij}^k$

$$
y_{ij}^k = \frac{u_{ij}^k (1 - u_{ij}^k) + (v_{ij}^k)^2}{1 - u_{ij}^k + v_{ij}^k} \tag{4}
$$

(4)　Calculate the exact value of expert $k$ triangular fuzzy judgment value $d_{ij}^k$

$$
d_{ij}^k = \min l_{ij}^k + y_{ij}^k \cdot \Delta_{\min}^{\max} \tag{5}
$$

(5)　Calculate the standard exact value of the evaluation by $p$ experts $d_{ij}$

$$
d_{ij} = \frac{1}{p} \cdot \sum_{k=1}^{p} d_{ij}^k \tag{6}
$$

(6)　Determine the fuzzy direct relationship matrix $D$.

$$
D = \begin{bmatrix}
0 & d_{12} & \cdots & d_{1n} \\
d_{21} & 0 & \ldots & d_{2n} \\
\vdots & \vdots & \vdots & \vdots \\
d_{n1} & d_{n2} & \cdots & 0
\end{bmatrix}
\tag{7}
$$

Step 4: calculate the normalized fuzzy direct relationship matrix $G$.

$$
G = (1 / \max_{1 \leq i \leq n} \sum_{j=1}^{n} d_{ij}) \cdot D \tag{8}
$$

Step 5: calculate the comprehensive impact matrix $T$.

$$
T = G(I - G)^{-1} \tag{9}
$$

where $I$ is the unit matrix.

Step 6: calculate influence degree $f_i$, influenced degree $e_i$, centrality degree $M_i$ and cause degree $N_i$.

$$
\begin{aligned}
f_i &= \sum_{j=1}^{n} t_{ij}, i = 1, 2, \cdots n \\
e_i &= \sum_{i=1}^{n} t_{ij}, j = 1, 2, \cdots n \\
M_i &= f_i + e_i, i = 1, 2, \cdots n \\
N_i &= f_i - e_i, i = 1, 2, \cdots n
\end{aligned}
\tag{10}
$$

where $t_{ij}$ is the elements of the comprehensive impact matrix $T$, representing the value of the impact of factor $i$ on factor $j$. $f_i$ is the sum of row values in matrix $T$, representing the comprehensive influence of factor $i$ on all other factors. $e_i$ is the sum of column values in matrix $T$, representing the comprehensive influence of all other factors on factor $i$. $M_i$ represents the importance of the factor in the system. $N_i$ represents the causal relationship between factors, $N_i > 0$ is the cause factor, and $N_i < 0$ is the effect factor.

*3.4. ANP Model*

ANP is a network analysis method based on the analytic hierarchy process (AHP) proposed by Saaty, a famous American operations researcher. This method improves the disadvantages of AHP, can consider the interdependence and influence between indexes, and can analyze many indexes which are coupled with each other. Finally, the weights of the indexes are determined by constructing the super matrix [45]. However, when constructing the ANP network structure model, it is difficult for experts to quantify the relationship between indexes based on their own experience, especially for the indirect impact between indexes as it is difficult to give an accurate evaluation. Therefore, this paper combines the comprehensive influence matrix of the DAMATEL method with the unweighted super matrix of the ANP method to construct the ANP network structure, calculate the mixed weights of each index, and then obtain the relative ranking of the indexes with high correlation degree and weight, which can make the evaluation results more objective and rigorous. In view of the complexity of the matrix operation involved in the ANP network, it is difficult to complete by manual calculation. In this paper, the super decision (SD) software specially applied to the ANP method is used for processing. Its basic steps are as follows [46,47].

Step 1: construct the network structure diagram.

Assume that $I_n$ is the $n$th group of elements and $F_{nm}$ is the $m$th factor in the $n$th group of elements. According to the relationship between evaluation indexes in the DEMATEL model, the ANP network structure is drawn with the actual situation. Where the arrows in the network structure indicate the mutual influence of indexes within 2 element groups, the 1-way arrows indicate that the indexes in the arrow tail element group influence the indexes in the head element group and the circular arrows indicate that the indexes within the element group influence each other.

Step 2: construct judgment matrix $B$.

Based on the constructed network structure, the importance of all the indexes in the network layer that have mutual influence relationships are compared. The 1–9 scale method proposed by Saaty is used to construct the judgment matrix $B$, and the scale of each index is determined as shown in Table 3.

$$
B = \begin{bmatrix}
b_{11} & \cdots & b_{1j} & \cdots & b_{1n} \\
\vdots & \vdots & \vdots & \vdots & \vdots \\
b_{i1} & \cdots & b_{ij} & \cdots & b_{in} \\
\vdots & \vdots & \vdots & \vdots & \vdots \\
b_{n1} & \cdots & b_{nj} & \cdots & b_{nn}
\end{bmatrix}
\tag{11}
$$

**Table 3.** Scale determined for each index in the judgment matrix.

| Scale | Meaning Description |
|---|---|
| 1 | Index *i* and index *j* have the same importance |
| 3 | Index *i* is slightly more important than index *j* |
| 5 | Index *i* is significantly more important than index *j* |
| 7 | Index *i* is extremely more important than index *j* |
| 9 | Index *i* is strongly more important than index *j* |
| 2, 4, 6, 8 | Intermediate value of the above adjacent judgment |
| Reciprocal | The importance ratio of index *i* to index *j* is one of the values above, then the importance ratio of index *j* to index *i* is its reciprocal |

The consistency test is performed for each judgment matrix. The test criteria are shown in Formula (12). If $CR \leq 0.1$, the judgment matrix passes the consistency test. If it does not pass the consistency test, the judgment matrix needs to be reconstructed until $CR \leq 0.1$.

$$CI = \frac{\lambda_{\max} - n}{n - 1}$$
$$CR = \frac{CI}{RI} \tag{12}$$

where $n$ represents the order of the judgment matrix, $\lambda_{\max}$ represents the maximum characteristic root of the matrix, $CI$ is the consistency index, $CR$ is the consistency ratio, and $RI$ is the random index. The comparison table of $RI$ values is shown in Table 4.

**Table 4.** Comparison table of random index (*RI*).

| *n* | 1 | 2 | 3 | 4 | 5 | 6 | 7 | 8 | 9 |
|---|---|---|---|---|---|---|---|---|---|
| *RI* | 0.00 | 0.00 | 0.58 | 0.90 | 1.12 | 1.24 | 1.32 | 1.41 | 1.45 |

Step 3: determine the unweighted super matrix *W*.

The unweighted super matrix can be obtained by integrating all judgment matrices that pass the consistency test, and its general form is shown below.

$$W = \begin{bmatrix} w_{11} & \cdots & w_{1j} & \cdots & w_{1n} \\ \vdots & \vdots & \vdots & \vdots & \vdots \\ w_{i1} & \cdots & w_{ij} & \cdots & w_{in} \\ \vdots & \vdots & \vdots & \vdots & \vdots \\ w_{n1} & \cdots & w_{nj} & \cdots & w_{nn} \end{bmatrix} \tag{13}$$

where $w_{ij}$ is the feature vector formed by the pairwise comparison of the elements in the *j*th factor group with the elements in the *i*th factor group. If the elements in the *j*-factor group have no effect on the elements in the *i*-factor group, then $w_{ij} = 0$.

Step 4: determine the weighted super matrix $\overline{W}$.

Each $w_{ij}$ in the unweighted super matrix is normalized, but the overall unweighted super matrix is not normalized. Therefore, the normalized weight judgment matrix of each dimension is multiplied with the unweighted super matrix to achieve the normalization of the super matrix, and thus the weighted super matrix is obtained.

$$\overline{W} = BW = \begin{bmatrix} b_{11}w_{11} & \cdots & b_{1j}w_{1j} & \cdots & b_{1n}w_{1n} \\ \vdots & \vdots & \vdots & \vdots & \vdots \\ b_{i1}w_{i1} & \cdots & b_{ij}w_{ij} & \cdots & b_{in}w_{in} \\ \vdots & \vdots & \vdots & \vdots & \vdots \\ b_{n1}w_{n1} & \cdots & b_{nj}w_{nj} & \cdots & b_{nn}w_{nn} \end{bmatrix} \tag{14}$$

Step 5: obtain the limit super matrix $\overline{W}^{\infty}$.

The calculation of the limit super matrix is an iterative, stabilization process that requires $n$ power operations on the weighted super matrix until the column vectors of the super matrix are identical, and the resulting matrix is the limit super matrix. When the limit value is convergent and unique, it is the obtained weight value of each index.

$$\overline{W}^{\infty} = \lim_{t \to \infty} (\overline{W})^{t} \tag{15}$$

*3.5. Mixed Weights*

In the weight distribution of the index system, sometimes the weight value of an index is not high, but its correlation value is high. Improving the ability of such indexes will improve the ability of other indexes, thus the level of the whole system will be improved. The correlation degree of each index is obtained using the FDEMATEL method, and the weight of each index is obtained using the ANP method. Combining the 2, the mixed weight of evaluation index is obtained using Formula (16). The mixed weight directly reflects the weight and influence degree of each index in the system, which can provide valuable index ranking results for decision makers.

$$Z = \overline{W}^{\infty} + T \times \overline{W}^{\infty} = (I + T)\overline{W}^{\infty} \tag{16}$$

## 4. Results

*4.1. Influence Relationship Determination Based on FDEMATEL*

According to the above index system and evaluation model, professors who have been engaged in intelligent mine construction or intelligent construction research for a long time and experts from relevant mining enterprises were invited to use the Delphi method to judge the strength of the role among the five first-grade indexes. The scoring results were shown in Appendix A. The average method was used to organize and summarize the survey results, Matlab2016a software was used to process the data, and finally the fuzzy direct relationship matrix $D$ was obtained using the CFCS method as follows.

$$D = \begin{array}{c} \\ I_1 \\ I_2 \\ I_3 \\ I_4 \\ I_5 \end{array} \begin{array}{c} I_1 \\ \begin{bmatrix} 0.000 \\ 0.172 \\ 0.089 \\ 0.100 \\ 0.236 \end{bmatrix} \end{array} \begin{array}{c} I_1 \\ 0.467 \\ 0.002 \\ 0.300 \\ 0.131 \\ 0.131 \end{array} \begin{array}{c} I_3 \\ 0.300 \\ 0.250 \\ 0.000 \\ 0.200 \\ 0.325 \end{array} \begin{array}{c} I_4 \\ 0.220 \\ 0.121 \\ 0.322 \\ 0.000 \\ 0.131 \end{array} \begin{array}{c} I_5 \\ 0.367 \\ 0.271 \\ 0.450 \\ 0.411 \\ 0.000 \end{bmatrix}$$

The comprehensive influence matrix $T$ was obtained from the fuzzy relationship matrix $D$ according to Formulas (8) and (9), as shown in the following matrix $T$. The influence degree $f_i$, the influenced degree $e_i$, the central degree $M_i$, and the cause degree $N_i$ of each first-grade index were calculated according to Formula (10). According to expert opinions and repeated tests, the threshold value is 0.300, i.e., the value below 0.300 in the comprehensive impact matrix means that the impact relationship between indexes is negligible. Then the values below 0.300 were adjusted to 0. The adjusted comprehensive impact matrix and calculation results are shown in Table 5.

$$T = \begin{array}{c} \\ I_1 \\ I_2 \\ I_3 \\ I_4 \\ I_5 \end{array} \begin{array}{c} I_1 \\ \begin{bmatrix} 0.350 \\ 0.338 \\ 0.362 \\ 0.301 \\ 0.384 \end{bmatrix} \end{array} \begin{array}{c} I_1 \\ 0.771 \\ 0.326 \\ 0.583 \\ 0.401 \\ 0.438 \end{array} \begin{array}{c} I_3 \\ 0.739 \\ 0.520 \\ 0.469 \\ 0.493 \\ 0.573 \end{array} \begin{array}{c} I_4 \\ 0.549 \\ 0.355 \\ 0.538 \\ 0.268 \\ 0.379 \end{array} \begin{array}{c} I_5 \\ 0.931 \\ 0.634 \\ 0.862 \\ 0.707 \\ 0.495 \end{bmatrix}$$

**Table 5.** Adjusted comprehensive impact matrix *T* and calculation results.

| First-Grade Indexes | $I_1$ | $I_2$ | $I_3$ | $I_4$ | $I_5$ | $f_i$ | $M_i$ | $N_i$ |
|---|---|---|---|---|---|---|---|---|
| $I_1$ | 0.350 | 0.771 | 0.739 | 0.549 | 0.931 | 3.340 | 5.075 | 1.605 |
| $I_2$ | 0.338 | 0.326 | 0.520 | 0.355 | 0.634 | 2.172 | 4.692 | −0.347 |
| $I_3$ | 0.362 | 0.583 | 0.469 | 0.538 | 0.862 | 2.814 | 5.608 | 0.021 |
| $I_4$ | 0.301 | 0.401 | 0.493 | 0.000 | 0.707 | 1.901 | 3.722 | 0.081 |
| $I_5$ | 0.384 | 0.438 | 0.573 | 0.379 | 0.495 | 2.270 | 5.899 | −1.359 |
| $e_i$ | 1.735 | 2.520 | 2.794 | 1.821 | 3.629 | - | - | - |

Based on the adjusted comprehensive impact matrix, the impact relationship between the first-grade evaluation indexes can be drawn in Figure 2. According to the dynamic influence relationship determined using the FDEMATEL method, the comprehensive influence matrix $T'$ of second-grade indexes and the degree of influence can be derived by following the above steps. Due to the large number of second-grade indexes, the impact matrix formed is relatively large, so only the calculation results of the impact degree of second-grade indexes are given in Table 6.

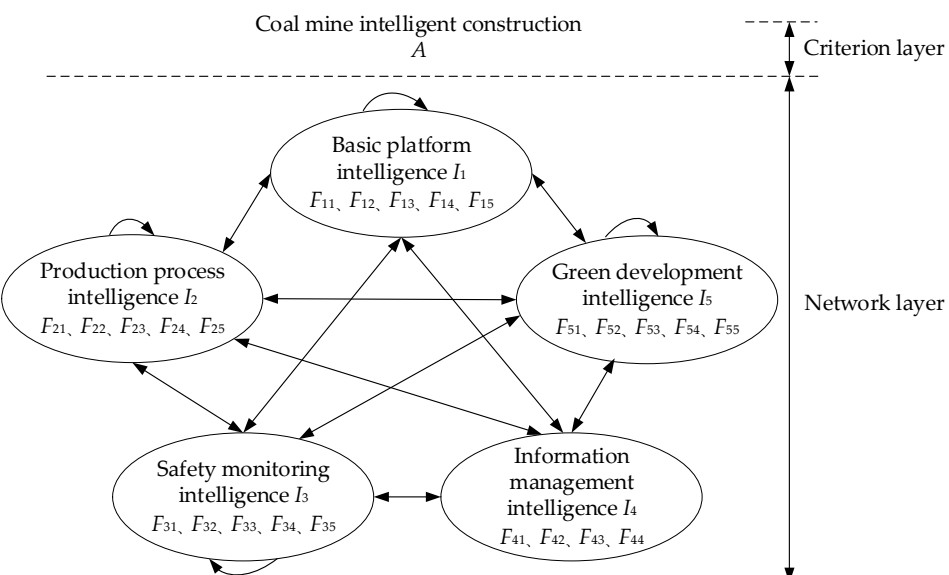

**Figure 2.** Influence relationship diagram of first-grade evaluation indexes.

**Table 6.** Calculation results of the influence degree of the second-grade indexes.

| Indexes | $f_i$ | $e_i$ | $M_i$ | $N_i$ |
|---|---|---|---|---|
| $F_{11}$ | 5.812 | 3.534 | 9.355 | 2.269 |
| $F_{12}$ | 5.694 | 3.669 | 9.364 | 2.025 |
| $F_{13}$ | 5.574 | 5.274 | 10.848 | 0.300 |
| $F_{14}$ | 5.964 | 4.471 | 10.435 | 1.494 |
| $F_{15}$ | 5.024 | 3.739 | 8.762 | 1.285 |
| $F_{21}$ | 4.662 | 4.260 | 8.921 | 0.402 |
| $F_{22}$ | 4.392 | 6.013 | 10.405 | −1.622 |
| $F_{23}$ | 4.028 | 4.754 | 8.781 | −0.726 |
| $F_{24}$ | 3.994 | 5.734 | 9.728 | −1.741 |
| $F_{25}$ | 5.512 | 2.666 | 8.179 | 2.846 |
| $F_{31}$ | 4.815 | 6.648 | 11.463 | −1.834 |
| $F_{32}$ | 4.635 | 6.247 | 10.882 | −1.613 |
| $F_{33}$ | 4.478 | 7.672 | 12.150 | −3.193 |
| $F_{34}$ | 5.610 | 3.766 | 9.376 | 1.845 |
| $F_{35}$ | 4.850 | 6.965 | 11.815 | −2.115 |

**Table 6.** *Cont.*

| Indexes | $f_i$ | $e_i$ | $M_i$ | $N_i$ |
|---|---|---|---|---|
| $F_{41}$ | 5.219 | 4.481 | 9.699 | 0.738 |
| $F_{42}$ | 5.562 | 5.050 | 10.612 | 0.512 |
| $F_{43}$ | 5.283 | 5.061 | 10.344 | 0.222 |
| $F_{44}$ | 5.277 | 4.772 | 10.048 | 0.505 |
| $F_{51}$ | 5.330 | 5.975 | 11.305 | −0.644 |
| $F_{52}$ | 4.750 | 4.557 | 9.307 | 0.194 |
| $F_{53}$ | 4.929 | 5.302 | 10.230 | −0.373 |
| $F_{54}$ | 4.579 | 5.105 | 9.684 | −0.525 |
| $F_{55}$ | 4.694 | 4.947 | 9.641 | −0.253 |

### *4.2. Index Weight Calculation Based on ANP*

Based on the ANP method, according to the relationship structure determined by the FDEMATEL, the ANP network model for intelligent construction of coal mines was constructed using super decision software, as shown in Figure 3. The group again invited professors who have long been engaged in intelligent mine construction or intelligent construction research and experts from relevant mining enterprises to make a two-by-two comparison of each evaluation index. Scoring was performed according to Table 2, thus obtaining the judgment matrix. First, the first-grade evaluation indexes were compared in pairs. Different judgment matrices can be formed according to the relationships between the first-grade evaluation indexes. Second, all second-grade evaluation indexes in each first-grade evaluation index were compared in pairs. Due to the large number of judgment matrices formed in the paper, only some representative judgment matrices are given for their detailed construction. Table 7 gives a comparison of the relevant first-grade evaluation indexes based on production process intelligence ($I_2$) as a criterion. Table 8 gives a comparison of each evaluation index within $I_4$ based on mobile internet construction ($F_{14}$) as a criterion. Table 9 gives a comparison of the relevant evaluation indexes within $I_5$ based on electrical equipment safety monitoring ($F_{23}$) as a criterion. In addition, the calculation of the judgment matrix needs to meet the consistency test, and the *CR* values in this paper are all less than 0.1, which meet the requirements.

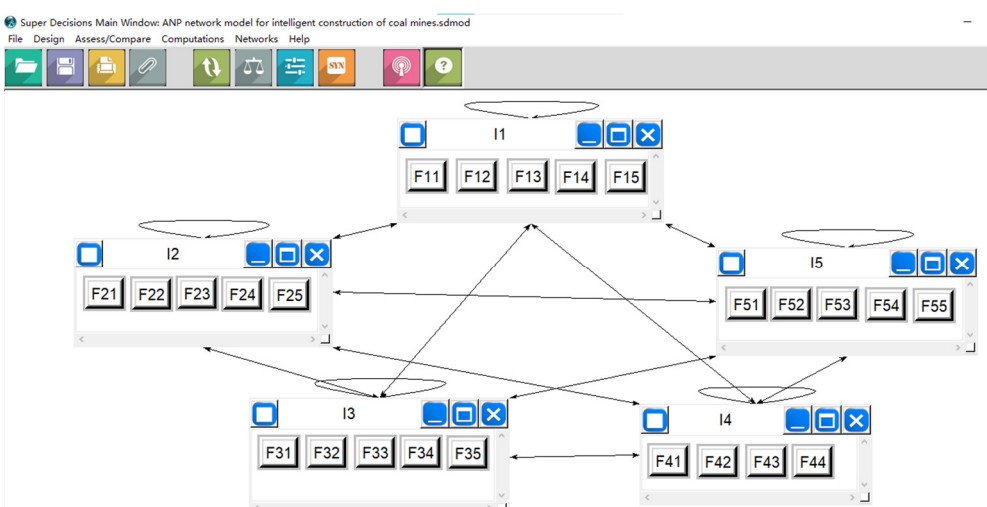

**Figure 3.** ANP network model for the intelligent construction of coal mines.

After inputting the constructed judgment matrix into super decision software in turn, the unweighted super matrix, weighted super matrix, and stable limit super matrix can be calculated. Due to the large order of each generated super matrix, only the final obtained stability limit super matrix column vectors are listed (the other matrices are not shown specifically, and those who need them can contact the authors).

$$\overline{W}^{\infty} = (0.084, 0.058, 0.058, 0.055, 0.061, 0.031, 0.023, 0.029, 0.020, 0.030, 0.065, 0.033,$$
$$0.059, 0.033, 0.045, 0.059, 0.056, 0.046, 0.053, 0.034, 0.018, 0.015, 0.012, 0.023)$$

**Table 7.** Judgment matrix of the first-grade evaluation index with $I_2$ as the criterion.

| $I_2$ | $I_1$ | $I_2$ | $I_3$ | $I_4$ | $I_5$ | Weight |
|---|---|---|---|---|---|---|
| $I_1$ | 1 | 1/2 | 1/3 | 1/2 | 3 | 0.134 |
| $I_2$ | 2 | 1 | 1/2 | 3 | 2 | 0.244 |
| $I_3$ | 3 | 2 | 1 | 4 | 3 | 0.395 |
| $I_4$ | 2 | 1/3 | 1/4 | 1 | 2 | 0.142 |
| $I_5$ | 1/3 | 1/2 | 1/3 | 1/2 | 1 | 0.085 |
| Consistency judgment | | | | $CR = 0.076 < 0.1$ | | |

**Table 8.** Judgment matrix of each evaluation index in $I_4$ with $F_{14}$ as the criterion.

| $F_{14}$ | $F_{41}$ | $F_{42}$ | $F_{43}$ | $F_{44}$ | Weight |
|---|---|---|---|---|---|
| $F_{41}$ | 1 | 3 | 2 | 1/2 | 0.285 |
| $F_{42}$ | 1/3 | 1 | 3 | 1/3 | 0.166 |
| $F_{43}$ | 1/2 | 1/3 | 1 | 1/4 | 0.097 |
| $F_{44}$ | 2 | 3 | 4 | 1 | 0.452 |
| Consistency judgment | | | $CR = 0.082 < 0.1$ | | |

**Table 9.** Judgment matrix of each evaluation index in $I_5$ with $F_{23}$ as the criterion.

| $F_{23}$ | $F_{51}$ | $F_{52}$ | $F_{53}$ | $F_{54}$ | $F_{55}$ | |
|---|---|---|---|---|---|---|
| $F_{51}$ | 1 | 3 | 1/2 | 4 | 3 | 0.282 |
| $F_{52}$ | 1/3 | 1 | 1/3 | 3 | 1/3 | 0.106 |
| $F_{23}$ | $F_{51}$ | $F_{52}$ | $F_{53}$ | $F_{54}$ | $F_{55}$ | |
| $F_{53}$ | 2 | 3 | 1 | 3 | 4 | 0.387 |
| $F_{54}$ | 1/4 | 1/3 | 1/3 | 1 | 1/3 | 0.065 |
| $F_{55}$ | 1/3 | 3 | 1/4 | 3 | 1 | 0.159 |
| Consistency judgment | | | $CR = 0.096 < 0.1$ | | | |

*4.3. Mixed Weight Calculation*

Based on the above limit super matrix and the comprehensive influence matrix of the second-grade indexes, the comprehensive weight of each second-grade evaluation index was calculated according to the mixed weight calculation Formula (16), using Matlab2016a software. Then the comprehensive weight of each second-grade index was ranked, and the weight of the first-grade index was calculated. Finally, the index weight table of coal mine intelligent construction evaluation system is formed, as shown in Table 10.

**Table 10.** Normalized results of evaluation index weight for intelligent construction of coal mine.

| First-Grade Indexes | Weight $Z$ | Second-Grade Indexes | Mixed Weight $Z$ | Sort |
|---|---|---|---|---|
| Basic platform intelligence $I_1$ | 0.247 | Database construction $F_{11}$ | 0.054 | 1 |
| | | Big data support $F_{12}$ | 0.049 | 3 |
| | | Model algorithm support $F_{13}$ | 0.048 | 4 |
| | | Mobile internet construction $F_{14}$ | 0.050 | 2 |
| | | Information safety system construction $F_{15}$ | 0.045 | 8 |

**Table 10.** *Cont.*

| First-Grade Indexes | Weight $Z$ | Second-Grade Indexes | Mixed Weight $Z$ | Sort |
|---|---|---|---|---|
| Production process intelligence $I_2$ | 0.177 | Reliability of working surface equipment $F_{21}$ | 0.037 | 17 |
| | | Intelligent diagnosis technology of equipment fault $F_{22}$ | 0.034 | 21 |
| | | Mining collaborative design $F_{23}$ | 0.032 | 23 |
| | | Production closed-loop control $F_{24}$ | 0.031 | 24 |
| | | Safety prevention closed-loop decision $F_{25}$ | 0.043 | 12 |
| Safety monitoring intelligence $I_3$ | 0.207 | Geological monitoring $F_{31}$ | 0.044 | 11 |
| | | Ventilation and fire safety monitoring $F_{32}$ | 0.037 | 16 |
| | | Electrical equipment safety monitoring $F_{33}$ | 0.041 | 15 |
| | | Personnel safety monitoring $F_{34}$ | 0.044 | 10 |
| | | Emergency rescue control $F_{35}$ | 0.041 | 14 |
| Information management intelligence $I_4$ | 0.184 | Information collection coverage capability $F_{41}$ | 0.046 | 6 |
| | | Data resource mining capability $F_{42}$ | 0.048 | 5 |
| | | Data statistical analysis ability $F_{43}$ | 0.044 | 9 |
| | | Information management system integration capability $F_{44}$ | 0.046 | 7 |
| Green development intelligence $I_5$ | 0.185 | Intelligent dust control $F_{51}$ | 0.042 | 13 |
| | | Intelligent control of toxic and hazardous substances $F_{52}$ | 0.036 | 20 |
| | | Drained water reuse $F_{53}$ | 0.037 | 18 |
| | | Clean energy utilization $F_{54}$ | 0.034 | 22 |
| | | Ecological restoration management $F_{55}$ | 0.036 | 19 |

## 5. Discussion

This study explores the indexes that affect the intelligent construction of coal mines by establishing a FDEMATEL-ANP evaluation model. The results of the study provide suggestions for promoting the intelligent construction of China's coal mines and ensuring the safe, efficient, economic, green, and sustainable development of the coal industry.

### 5.1. Comparative Validation

This subsection presents the results of the comparative analysis of the FDEMATEL-ANP comprehensive evaluation with the ANP method and the SWARA method to verify the applicability and scientific validity of the FDEMATEL-ANP model.

SWARA is a new multi-criteria decision-making method (MCDM) capable of assessing the accuracy of experts regarding the weighting criteria in the methodology process. In this method, the highest priority will be assigned to the most valuable criterion and the lowest priority will be assigned to the least valuable evaluation criterion. The basic steps can be found in Refs. [48–50]. Due to the limitation of space, this paper takes the first-grade index $I_1$ as an example and invites five experts of the subject group to calculate the weight values of the second-grade indexes $F_{11}{\sim}F_{15}$ according to the operation steps of SWARA method, and the results are shown in Table 11.

**Table 11.** Calculation results of index weights of SWARA method.

| Index Code | Expert 1 | Expert 2 | Expert 3 | Expert 4 | Expert 5 | Average Value |
|---|---|---|---|---|---|---|
| $F_{11}$ | 0.235 | 0.174 | 0.111 | 0.059 | 0.302 | 0.176 |
| $F_{12}$ | 0.059 | 0.140 | 0.093 | 0.135 | 0.038 | 0.093 |
| $F_{13}$ | 0.035 | 0.060 | 0.074 | 0.209 | 0.077 | 0.091 |
| $F_{14}$ | 0.139 | 0.038 | 0.139 | 0.345 | 0.195 | 0.171 |
| $F_{15}$ | 0.116 | 0.236 | 0.031 | 0.029 | 0.028 | 0.088 |

Then, the weight values of the second-grade indexes $F_{11}$ to $F_{15}$ calculated by FDEMATEL-ANP, ANP, and SWARA were compared and analyzed. The results of the ranking of the obtained index weight values are shown in Table 12.

**Table 12.** Ranking of weight values of indexes $F_{11} \sim F_{15}$.

| Applied Methods | $F_{11}$ | $F_{12}$ | $F_{13}$ | $F_{14}$ | $F_{15}$ |
|:---:|:---:|:---:|:---:|:---:|:---:|
| ANP | 0.084 | 0.058 | 0.058 | 0.055 | 0.061 |
| Ranking | (1) | (3) | (4) | (5) | (2) |
| FDEMATEL-ANP | 0.054 | 0.049 | 0.048 | 0.050 | 0.045 |
| Ranking | (1) | (3) | (4) | (2) | (5) |
| SWARA | 0.176 | 0.093 | 0.091 | 0.171 | 0.088 |
| Ranking | (1) | (3) | (4) | (2) | (5) |

According to the results listed in Table 12, a comparative analysis of the FDEMATEL-ANP model and the ANP method revealed that the weight values of the first three indexes were ranked in the same order, but the weight values of the last two indexes were ranked differently. This discrepancy occurs because the FDEMATEL-ANP model uses the DEMATEL method to determine the causal relationships between the indexes, identifies the network association structure of the indexes, and then uses the ANP method to calculate the weight values for each index. The process also uses triangular fuzzy numbers to reduce the variability of subjective expert judgments. However, the ANP method alone cannot fully consider the correlation and interaction between indexes, and the calculation results are highly subjective.

By comparing and analyzing the calculation results of the FDEMATEL-ANP model with those of the SWARA method, we can see that the calculation results of both are in the same order. It is shown that the constructed FDEMATEL-ANP model has strong applicability to the evaluation study of coal mine intelligent construction. Since the calculation of index weights is carried out, there exists a certain index with a low weight value, but its correlation is high. Compared with the SWARA method, the mixed weights calculated by the FDEMATEL-ANP model can not only obtain scientific index weight values, but also take into account the logical relationships between the indexes, attribute characteristics, and finally can determine the indexes with higher correlations and weight values.

*5.2. Results Analysis*

(1) Analysis of index influence relationship. From the calculation results of the FDEMATEL method, it can be seen that the influence degree of the basic platform intelligence ($I_1$) is 3.340, which is the highest in the ranking of the influence degrees in the first-grade index, and the influence degrees in the other first-grade indexes are less than three. This shows that the basic platform intelligence is an important cornerstone of the intelligent construction of coal mines, which can provide comprehensive services for the construction of various business systems at a later stage and ensure reliable data collection, transmission, storage, and application. The influence degrees of mobile internet construction ($F_{14}$), database construction ($F_{11}$), and big data support ($F_{12}$) are ranked in the top three in all second-grade indexes, and these three indexes belong to the basic platform intelligence ($I_1$) in the first-grade indexes. Once again, it confirms the key influence of the intelligent construction of the basic platform in promoting the process of intelligent coal mining. Green development intelligence ($I_5$) is the most influenced index among the first-grade indexes, with an influence degree of 3.629, and the influence degrees of other first-grade indexes are less than three. It shows that coal mining should pay attention to the green service of the whole life cycle, to achieve the essential green of coal mining, in order to reach the green development goal of the intelligent construction of coal mines. The top three influenced second-grade indexes are electrical equipment safety monitoring ($F_{33}$), emergency rescue control ($F_{35}$), and geological monitoring ($F_{31}$), and these three indexes belong to the first-grade indexes of safety monitoring intelligence ($I_3$). It shows that the services and applications of safety monitoring are based on the collection of real-time and historical data related to coal mine production scenarios and industrial scenarios, which are realized by integrating and analyzing the collected data.

(2)  Analysis of index attributes. The FDEMATEL method analysis shows that the basic platform intelligence ($I_1$), safety monitoring intelligence ($I_3$), and information management intelligence ($I_4$) are the cause factors, and the production process intelligence ($I_2$) and green development intelligence ($I_5$) are the result factors in the first-grade indexes. This indicates that the most important thing to consider in the intelligent construction of coal mines is the application of information technology and the development of safe production. In the whole intelligent coal mine construction evaluation index system, there are 13 second-grade indexes with positive cause degrees, which are cause factors and easily influence other factors in the system. Among them, safety prevention closed-loop decisions ($F_{25}$), database construction ($F_{11}$), and big data support ($F_{12}$) are strong cause factors, and their cause degree is 2.846, 2.269, and 2.025, respectively, while the cause degree of other cause factors is less than two. It indicates that the strong cause factors have an important influence on promoting the intelligent construction of coal mines and should be focused on in the intelligent construction of coal mines. The cause degrees of the remaining 11 second-grade indexes are negative. These indexes are result factors, which are easily influenced by the above-mentioned cause factors in the system, and focusing on the changes in these factors can clarify the improvement path and development direction of coal mine intelligent construction. Among the result factors, electrical equipment safety monitoring ($F_{33}$) and emergency rescue control ($F_{35}$) are strong result factors with a cause degree of $-3.193$ and $-2.115$, respectively, while the cause degree of other result factors is greater than $-2$. It indicates that real-time online monitoring of coal mine hazards, determination of risk types, levels, and corresponding solutions through the application of information technology, such as big data, the internet of things, and artificial intelligence, are effective ways to achieve intelligent coal mine safety production and management.

(3)  Analysis of index importance. Mixed weights are the comprehensive influence weights obtained by considering the interaction between the evaluation indexes and combining the ANP method. It reflects the weight and correlation degree of each element, provides the priority order of each element, and its ranking result is the ranking of each evaluation index in the whole coal mine intelligent construction evaluation system. Therefore, the best way to promote the intelligent construction of coal mines is to improve the evaluation indexes that have greater weight and relevance in the whole evaluation system. Improving such indexes will increase the value of other evaluation indexes and can promote the high-quality development of the whole intelligent construction of coal mines. From the mixed weights in Table 10, it can be seen that the relative importance of the first-grade indexes is ranked as $I_1 > I_3 > I_5 > I_4 > I_2$, the highest relative importance is the basic platform intelligence ($I_1$), and the lowest is the production process intelligence ($I_2$). This is because the information technology system provides technology and equipment support for the intelligent construction of coal mines, which is an important cornerstone of the intelligent construction of coal mines. Therefore, coal mine intelligent construction enterprise managers need to pay more attention to the intelligent construction of the basic platform. The lowest weight of production process intelligence ($I_2$) does not mean that the construction of production process intelligence is insignificant, but it is the least important compared with other first-grade indexes, and it is still an important goal to promote coal mining enterprises to realize a reduction in field staff and efficient development of resources. In the mixed weight ranking of second-grade indexes, database construction ($F_{11}$), mobile internet construction ($F_{14}$), big data support ($F_{12}$), and model algorithm support ($F_{13}$) are the top four, and all of them belong to the first-grade index base platform intelligence ($I_1$), so the base platform intelligence will provide the basic guarantee for the construction of various business systems in the coal mine.

*5.3. Countermeasures and Suggestions to Promote the Intelligent Construction of Coal Mines*

At present, the intelligent construction of China's coal mines is in its infancy, and although the state has issued some standards and guidelines for different provinces and regions, the geological conditions of the mines are different, the mining conditions vary widely, the mining process varies, and the specific design and acceptance of the mines still lack a unified standard document. Due to the large number of index factors affecting the intelligent construction of coal mines, there are complex and non-linear interactions between various factors. In order to accurately assess the level of intelligence in a coal mine, it is necessary to consider information on the basic parameters of the coal mine and to perform dynamic assessments based on different coal mine conditions to improve the reliability of the results. Therefore, the following countermeasures and suggestions are proposed to promote the intelligent construction of coal mines.

(1)　Refine the top-level design of the industry and form a unified construction standard. In promoting the construction of intelligent mines, we should further refine the construction standards and standardize the standards of various intelligent parameters, interfaces, and protocols to lay a good foundation for later unified management. At the same time, promote the technical support and security forces, promote teaching and research forces integration, and strengthen long-term stable cooperation with well-known universities, research institutes, and leading information technology enterprises at home and abroad in related fields, to make up for the lack of their own strength.

(2)　Strengthen the integration of advanced technologies and accelerate the process of mine intelligence. Promote the integration of cloud computing, big data, the internet of things, artificial intelligence, mobile internet, and other new generation information technology with intelligent mines in depth, and promote the integration of cross-disciplinary technologies.

(3)　Strengthen the cultivation of professional talents and build innovative technical teams. Strengthen academic exchanges within the industry, vigorously promote cooperation between mining enterprises and relevant research institutes, and support the application and transformation of scientific and technological achievements. Continuously improve the ability of mine managers and technicians to use data to analyze problems and solve them, strengthen training and education on the quality of information technology for all staff, and build a composite and innovative core talent team.

*5.4. Limitations and Future Research*

In this paper, through literature research, expert interviews, and questionnaire surveys, the factors affecting the intelligent construction of coal mines are determined and the evaluation index system of intelligent construction of coal mines is established. A combination of FDEMATEL and ANP methods was used to determine the index weights, which improved the deficiencies arising from the single-method assignment in existing studies. However, this paper still has some limitations and future research will be conducted on the following. (1) Complementary research methods. Combined with the ISM method to divide the hierarchy of influencing factors in the system, the system with complex relationships is decomposed into a number of concise and clear subsystems, and the role of the relationship between factors is further analyzed. (2) Optimization of the evaluation model. In the evaluation model constructed in this paper, both DEMATEL and ANP are subjective assignment methods, and the index weights are mainly determined using the knowledge and experience of decision makers or experts in related fields, which overly rely on the subjective consciousness of evaluation experts and will cause the evaluation results to be incomplete to different degrees. At a later stage, we consider combining objective assignment methods such as principal component analysis (PCA) and the entropy weight method (EWM) to optimize the evaluation results. The factors affecting the intelligent construction of coal mines are ranked with the help of the TOPSIS method. (3) Improvement of evaluation index system. As the study of intelligent construction in coal mines is a complex

issue, there are more factors affecting the degree of intelligent construction. In the future, we will consider analyzing and identifying influencing factors from more dimensions to further improve the evaluation index system. For example, the economic level, policy level, and talent pool level of coal mine intelligent construction. It will also conduct multi-area data collection, multi-case analysis, and simulation to establish a dynamic database of coal mine intelligent construction evaluation indexes.

## 6. Conclusions

This study uses the data obtained by expert consultation methods and questionnaire survey methods as the source of research data and constructs the index system affecting the intelligent construction of coal mines based on literature research, relevant policy documents, and engineering cases. Combined with the FDEMATEL-ANP model, the mechanism of intelligent construction in coal mines was studied. The main conclusions are as follows:

(1) The factors influencing the intelligent construction of coal mines are summarized. Through literature research, expert consultation, and a questionnaire survey, the evaluation index system of coal mine intelligent construction was constructed. It includes 5 first-grade indexes and 24 second-grade indexes.

(2) The FDEMATEL-ANP evaluation model was applied to analyze the comprehensive impact of various influencing factors on the intelligent construction of coal mines. The results show that safety prevention closed-loop decisions, database construction, and big data support have a significant impact on the intelligent construction of coal mines. Database construction, mobile internet construction, big data support, and model algorithm support are the key factors affecting the intelligent construction of coal mines.

(3) By analyzing and constructing the index system of influencing factors on coal mine intelligent construction, this study can effectively guide the intelligent construction of coal mines. The case verification shows that the comprehensive weights of indexes calculated by the FDEMATEL-ANP model are basically consistent with the results calculated by the ANP model and consistent with the results calculated by the SWARA model. This calculation result matches with the actual situation and verifies the scientific nature and applicability of the evaluation model. Based on this, the coal mining industry and related enterprises can improve the corresponding standards and take corresponding measures to promote the high-quality development of intelligent coal mine construction.

**Author Contributions:** Conceptualization, D.Y. and L.R.; methodology, L.H. and M.H.; investigation, W.Z. and J.T.; writing—original draft preparation, L.H.; writing—review and editing, D.Y. All authors have read and agreed to the published version of the manuscript.

**Funding:** This research was funded by the National Natural Science Foundation of China (grant number U1810203).

**Institutional Review Board Statement:** Not applicable.

**Informed Consent Statement:** Not applicable.

**Data Availability Statement:** Not applicable.

**Acknowledgments:** The authors wish to acknowledge the financial support for this research by the National Fund Committee, the experts involved in the interview, and the coal mine enterprise manager who participated in the questionnaire, as well as the editors and reviewers.

**Conflicts of Interest:** The authors declare no conflict of interest.

## Appendix A

**Table A1.** Expert 1's rating of the first-grade indexes.

| First-Grade Indexes | $I_1$ | $I_2$ | $I_3$ | $I_4$ | $I_5$ |
|---|---|---|---|---|---|
| $I_1$ | 0 | 4 | 4 | 1 | 3 |
| $I_2$ | 5 | 0 | 4 | 3 | 5 |
| $I_3$ | 2 | 3 | 0 | 2 | 2 |
| $I_4$ | 4 | 2 | 1 | 0 | 1 |
| $I_5$ | 5 | 3 | 3 | 2 | 0 |

**Table A2.** Expert 2's rating of the first-grade indexes.

| First-Grade Indexes | $I_1$ | $I_2$ | $I_3$ | $I_4$ | $I_5$ |
|---|---|---|---|---|---|
| $I_1$ | 0 | 1 | 3 | 1 | 1 |
| $I_2$ | 1 | 0 | 2 | 1 | 3 |
| $I_3$ | 2 | 2 | 0 | 2 | 2 |
| $I_4$ | 3 | 2 | 1 | 0 | 1 |
| $I_5$ | 2 | 4 | 2 | 4 | 0 |

**Table A3.** Expert 3's rating of the first-grade indexes.

| First-Grade Indexes | $I_1$ | $I_2$ | $I_3$ | $I_4$ | $I_5$ |
|---|---|---|---|---|---|
| $I_1$ | 0 | 1 | 3 | 4 | 5 |
| $I_2$ | 1 | 0 | 3 | 1 | 4 |
| $I_3$ | 3 | 3 | 0 | 2 | 4 |
| $I_4$ | 1 | 3 | 4 | 0 | 5 |
| $I_5$ | 1 | 4 | 4 | 5 | 0 |

**Table A4.** Expert 4's rating of the first-grade indexes.

| First-Grade Indexes | $I_1$ | $I_2$ | $I_3$ | $I_4$ | $I_5$ |
|---|---|---|---|---|---|
| $I_1$ | 0 | 1 | 1 | 1 | 4 |
| $I_2$ | 1 | 0 | 1 | 2 | 3 |
| $I_3$ | 2 | 2 | 0 | 2 | 2 |
| $I_4$ | 1 | 1 | 1 | 0 | 1 |
| $I_5$ | 1 | 3 | 1 | 4 | 0 |

**Table A5.** Expert 5's rating of the first-grade indexes.

| First-Grade Indexes | $I_1$ | $I_2$ | $I_3$ | $I_4$ | $I_5$ |
|---|---|---|---|---|---|
| $I_1$ | 0 | 5 | 1 | 5 | 1 |
| $I_2$ | 1 | 0 | 1 | 2 | 1 |
| $I_3$ | 2 | 2 | 0 | 3 | 2 |
| $I_4$ | 1 | 1 | 1 | 0 | 3 |
| $I_5$ | 1 | 3 | 1 | 1 | 0 |

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
