# Peer review of "Evaluation Model Research of Coal Mine Intelligent Construction Based on FDEMATEL-ANP"

_sustainability, doi:10.3390/su15032238_

Round 1

Reviewer 1 Report

Major Revision:

1.      The novelty of the paper is not clearly defined. Clearly define your contribution in relation to previous work.

2.      The main disadvantage of this paper is that there is no comparative analysis with state-of-the-art methods. You should compare your results with at least three state-of-the-art methods.

3.      The proposed method is also not clearly defined.

4.      The authors have not defined their future work or study in the conclusions section.

Minor revision:

1.      Reference number 3 has multiple authors, so instead of writing just "Fu," you should write "Fu et al." Please make sure to correct any other references in the manuscript that follow this pattern.

Author Response

Point 1: The novelty of the paper is not clearly defined. Clearly define your contribution in relation to previous work.

Response 1: The innovations and the work done on this paper have been added and revised. The innovation of this paper is clearly stated at the end of the introduction and the literature review. The innovation is reflected in the fact that the current research on intelligent mine construction mainly focuses on the review of development prospects, key technologies and construction paths, and only a few scholars have been involved in the evaluation study of intelligent coal mine construction. To complement the theoretical research in this area, the following work has been done in this paper.

(1) In this paper, according to the content and characteristics of intelligent construction of coal mines, combined with the relevant policies and regulations of intelligent construction of coal mines, the evaluation index system of intelligent construction of coal mines was established through literature review, expert interview method and questionnaire survey method.

(2) Taking into full consideration the advantages and disadvantages of each research method, we propose to construct an evaluation model for intelligent construction of coal mines based on FDEMATEL-ANP.

(3) Calculate the comprehensive ranking of each evaluation index and find out the key indexes to promote the intelligent construction of coal mines.

Point 2: The main disadvantage of this paper is that there is no comparative analysis with state-of-the-art methods. You should compare your results with at least three state-of-the-art methods.

Response 2: This paper is modified to introduce the SWARA method in the multi-criteria decision-making method (MCDM) in subsection 5.1. Taking the first-grade index I1 as an example, the comprehensive evaluation results of FDEMATEL-ANP are compared and analyzed with ANP method and SWARA method to verify the applicability and scientific validity of FDEMATEL-ANP model.

Point 3: The proposed method is also not clearly defined.

Response 3: After the revision of this paper, the definition, applicability, operation steps, and calculation process of the used fuzzy set theory, DEMATEL method, and ANP method are explained in detail in Section 3. It also provides some explanation of the definitions related to SWARA method in the chapter of comparative analysis of methods.

Point 4: The authors have not defined their future work or study in the conclusions section.

Response 4: Due to the confusing organization of the first draft, the limitations of this paper and the content of future research work are not well expressed. After the revision, the limitations of this paper and the future research work planned are detailed in subsection 5.4 of the paper.

Point 5: Reference number 3 has multiple authors, so instead of writing just "Fu," you should write "Fu et al." Please make sure to correct any other references in the manuscript that follow this pattern.

Response 5: The format of the full-text references has been checked and changes have been made to those that do not meet the requirements.

Reviewer 2 Report

Dear Authors, my best X-Mass wishes!

The article is very relevant and interesting for readers in the field. However, some notes are quite serious, in my opinion, should be revised before promotion:

1. The structure of the manuscript is not optimal; there is no Discussion section where your advances in the field are shown. Also in this section should bear the advantages over existing researches that your team are achieved. In addition, there is no Results section, where your findings are given and explain in brief. 

2. The References list should be more international.

3. The Conclusion should contain methodological advances of yours.

Good luck!

Author Response

Point 1: The structure of the manuscript is not optimal; there is no Discussion section where your advances in the field are shown. Also in this section should bear the advantages over existing researches that your team are achieved. In addition, there is no Results section, where your findings are given and explain in brief.

 Response 1: The full text has been reorganized and Section 5 has been revised as a discussion section. The SWARA method in the multi-criteria decision-making method (MCDM) has been introduced in subsection 5.1. The FDEMATEL-ANP comprehensive evaluation results were compared and analyzed with ANP method and SWARA method to verify the applicability and scientific validity of the FDEMATEL-ANP model. And the analysis of the results of the evaluation study of this paper is carried out in subsection 5.2.

Point 2: The References list should be more international.

Response 2: References have been revised in the text and more international literature has been added.

Point 3: The Conclusion should contain methodological advances of yours.

Response 3: The full text has been reorganized and the content of the Conclusion has been revised. The first is the addition of a discussion in Section 5, which contrasts and analyzes the scientific validity and merits of the methods used in this paper. Second, the methodological soundness of this paper is also highlighted in the Conclusion.

Reviewer 3 Report

The paper proposes an intelligent coal mine construction evaluation model that combines two subjective weighting methods such as Fuzzy Decision-Making Trial and Evaluation Laboratory (FDEMATEL) and Analytical Network Process (ANP). It is finely described and correctly presented. This paper has a potential to be of interest in the mining industry, especially in the environmental protection caused by mining activities. However, there are some critical shortcomings that should be revised and improved.

 1.       First, the literature review related to MCDM methods is quite weak. Since DEMATEL and ANP belong to the group of subjective weighting methods, studies that apply some of the methods from that group such as the FUCOM method, the SWARA method, the PIPRECIA method and other methods should be added.

2.       Creation the intelligent construction of the coal mine is very complex and difficult task for mining engineers. Coal mining companies tend to achieve as much as possible positive final financial plan, depending on the fluctuations of the coal price on the market. Especially in the China, where coal is still a vital mineral resource of the energy structure. You have emphasized the significant factors” (in your case “indexes”) that influence the creation of the intelligent construction of the coal mine but you have not considered the most dominant factor, which is the economic component of the coal mine. That component (coal price) is actually the most important indicator of coal mine optimization and represents one of the key benchmark in the process of intelligent construction. In addition to these 5 first-grade indexes, your developed evaluation model would be really powerful by involving another index related to the economic characteristics of intelligent coal mine construction.

3.       Comparative analysis and sensitivity analysis of the proposed evaluation model should be added in the paper. It would be of great importance for the verification and validation of the proposed evaluation model as well as the obtained results. I recommend you to create a comparative analysis between your proposed methods with the above-mentioned subjective weighting methods such as FUCOM method, SWARA method and PIPRECIA method. Certainly, other subjective weighting methods can be included in the comparative analysis with respect to your proposed methods. What are advantages and disadvantages of the proposed methods comparing the other methods? Please give a detailed discussion of the obtained results from the comparative analysis.

4.       The formula marks are missing in the line 358 and line 367. Please revise them.

5.       In Section 5.1, the Initial matrix with input data on the basis of which you calculate the weights of each index is also missing. Please add it. It can be very useful for the purpose of educating young researchers to better understand this calculation process.

6. Phrase the Conclusions section. The authors only repeat the steps of the evaluation model of coal mine intelligent construction, but the findings of the study are not clearly stated. Conclusions must go deeper, it would be more interesting if the authors focus more on the significance of their findings regarding the importance of the interrelationship between the obtained results and sustainable development/innovative processes in the mining industry and the barriers to do it, what would be the consequences, in the real coal mine, in changing the observed situation and what would be the ways, in the real world, to change/improve the perceived situation. Theoretical contributions of this paper, limitations of the proposed method as well as case study limitations should be given in the conclusion section. Are you considering the possibility of implementing some objective weighting methods in the future works or creating a combined (hybrid) approaches between your applied subjective and well-known objective weighting methods? What are your suggestions for future research and how is the proposed method useful for solving the real-life problems?

Author Response

Point 1: First, the literature review related to MCDM methods is quite weak. Since DEMATEL and ANP belong to the group of subjective weighting methods, studies that apply some of the methods from that group such as the FUCOM method, the SWARA method, the PIPRECIA method and other methods should be added.

 Response 1: A summary of the MCDM approach has been added to the literature review section in Section 2, and the added literature is focused on research in the mining and energy fields. And the SWARA method is introduced in the discussion section of Section 5, which adds a comparative analysis of the methods. Taking the first-grade index I1 as an example, the comprehensive evaluation results of FDEMATEL-ANP are compared and analyzed with ANP method and SWARA method to verify the applicability and scientific validity of FDEMATEL-ANP model.

Point 2: Creation the intelligent construction of the coal mine is very complex and difficult task for mining engineers. Coal mining companies tend to achieve as much as possible positive final financial plan, depending on the fluctuations of the coal price on the market. Especially in the China, where coal is still a vital mineral resource of the energy structure. You have emphasized the significant “factors” (in your case “indexes”) that influence the creation of the intelligent construction of the coal mine but you have not considered the most dominant factor, which is the economic component of the coal mine. That component (coal price) is actually the most important indicator of coal mine optimization and represents one of the key benchmark in the process of intelligent construction. In addition to these 5 first-grade indexes, your developed evaluation model would be really powerful by involving another index related to the economic characteristics of intelligent coal mine construction.

Response 2: After your guidance, I realized the limitations of the constructed evaluation model. However, considering that the evaluation model is the basis of the research results and conclusions, if the evaluation indexes are readjusted, then all the questionnaires, expert interviews and calculations will be overturned and repeated, which will bring great difficulties to the revision of this paper. Based on this, the limitations of this paper and future research plans are added in Section 5, expecting to further improve the evaluation model with more rigorous scientific research in future studies.

Point 3: Comparative analysis and sensitivity analysis of the proposed evaluation model should be added in the paper. It would be of great importance for the verification and validation of the proposed evaluation model as well as the obtained results. I recommend you to create a comparative analysis between your proposed methods with the above-mentioned subjective weighting methods such as FUCOM method, SWARA method and PIPRECIA method. Certainly, other subjective weighting methods can be included in the comparative analysis with respect to your proposed methods. What are advantages and disadvantages of the proposed methods comparing the other methods? Please give a detailed discussion of the obtained results from the comparative analysis.

Response 3: Taking into account your review comments, the structure of the article has been reorganized. Firstly, a discussion section is added in Section 5, and a comparative analysis of the proposed evaluation models is presented in Subsection 5.1. Secondly, the SWARA method is introduced in this paper, and the FDEMATEL-ANP comprehensive evaluation results are compared and analyzed with the ANP method and SWARA method to verify the applicability and scientificity of the FDEMATEL-ANP model, taking the first-grade index I1 as an example. The advantages and disadvantages of the evaluation model and the results obtained from the comparative analysis are also detailed in the discussion section.

Point 4: The formula marks are missing in the line 358 and line 367. Please revise them.

Response 4: The missing formula marks have been added in the text and the full content has been rechecked.

Point 5: In Section 5.1, the Initial matrix with input data on the basis of which you calculate the weights of each index is also missing. Please add it. It can be very useful for the purpose of educating young researchers to better understand this calculation process.

Response 5: Since the initial matrix of expert scores occupies more space in the text, it has a certain degree of influence on the structure of the full text. The initial matrix is therefore supplemented in the appendix, as described in Appendix A.

Point 6: Phrase the Conclusions section. The authors only repeat the steps of the evaluation model of coal mine intelligent construction, but the findings of the study are not clearly stated. Conclusions must go deeper, it would be more interesting if the authors focus more on the significance of their findings regarding the importance of the interrelationship between the obtained results and sustainable development/innovative processes in the mining industry and the barriers to do it, what would be the consequences, in the real coal mine, in changing the observed situation and what would be the ways, in the real world, to change/improve the perceived situation. Theoretical contributions of this paper, limitations of the proposed method as well as case study limitations should be given in the conclusion section. Are you considering the possibility of implementing some objective weighting methods in the future works or creating a combined (hybrid) approaches between your applied subjective and well-known objective weighting methods? What are your suggestions for future research and how is the proposed method useful for solving the real-life problems?

Response 6: The content of the conclusion has been recapitulated in the text. The results of this paper identify the factors that have a significant impact on the intelligent construction of coal mines, and determine the key factors to promote the intelligent construction of coal mines. The research results are important for the coal mining industry and enterprises to improve the standards of intelligent mine construction and promote the high-quality development of intelligent coal mine construction. After reorganizing the structure of the article, the discussion in Section 5 was added to the text, and countermeasures and suggestions for promoting the intelligent construction of coal mines were added in Subsection 5.3. Meanwhile, the limitations of the study and plans for future research are added in subsection 5.4. In view of the limitations of this paper's research, the future will mainly focus on the supplementation of research methods, the optimization of evaluation models, and the improvement of evaluation index systems, which can be seen in the paper.

Round 2

Reviewer 1 Report

All of the comments have been answered correctly and no further modifications are needed.

Reviewer 2 Report

Good luck!

Reviewer 3 Report

The authors have successfully addressed most, but not all, of the highlighted shortcomings and in my opinion have improved the paper decently. The paper is now suitable for publication in “Sustainability”.